

# Seismic detection of rockslides at regional scale: Examples from the Eastern Alps and feasibility of kurtosis-based event location

Florian Fuchs[1], Wolfgang Lenhardt[2], Götz Bokelmann[1], and the AlpArray Working Group[*]

[1]Department of Meteorology and Geophysics, University of Vienna, Althanstraße 14, UZA 2, 1090 Vienna, Austria
[2]Central Institute for Meteorology and Geodynamics, ZAMG, Vienna, Austria
[*]A full list of authors and their affiliations appears at the end of the paper.

*Correspondence to:* Florian Fuchs (florian.fuchs@univie.ac.at)

**Abstract.** Seismic records can provide detailed insight into the mechanisms of gravitational mass movements. Catastrophic events that generate long-period seismic radiation have been studied in detail, and monitoring systems have been developed for applications on very local scale. Here we demonstrate that similar techniques can also be applied to regional seismic networks which show great potential for real-time and large-scale monitoring and analysis of rockslide activity. This manuscript studies 21 moderate-size to large rockslides in the Eastern Alps that were recorded by regional seismic networks within distances of few tens of kilometers to more than 200 km. We develop a simple and fully automatic processing chain that detects, locates, and classifies rockslides based on vertical-component seismic records. We show that a kurtosis-based onset picker is suitable to detect the very emergent onsets of rockslide signals, and to locate the rockslides within a few kilometers from the true origin, using a grid search and a 1D seismic velocity model. Automatic discrimination between rockslides and local earthquakes is possible by a combination of characteristic parameters extracted from the seismic records, such as kurtosis or maximum-to-mean amplitude ratios. We attempt to relate the amplitude of the seismic records with the documented rockslide volume and reveal a potential power-law in agreement with earlier studies. Since our approach is based on simplified methods we suggest and discuss how each step of the automatic processing could be expanded and improved to achieve more detailed results in the future.

## 1 Introduction

Gravitational mass movements shape the surface of our planet and pose sincere hazards to human population, in particular in densely populated mountain regions, such as the European Alps. Understanding the triggers of slope failures allows to better evaluate their impact on the evolution of geomorphology and to design mitigation measures or early warning systems. However, such events may occur spontaneously and in remote areas and thus remain undetected in many cases. This can introduce significant uncertainty to e.g. event inventories and triggering studies. Yet, comprehensive knowledge and reliable event data are of particular importance for the assessment of hazards imposed by rapid gravitational mass movements (Petschko et al., 2014; Lima et al., 2017). This renders remote and preferably real-time detection methods for rapid gravitational mass movements highly desirable. Classical approaches such as remote sensing via satellite imagery or stationary slope monitoring systems are usually limited in either temporal or spatial resolution and cannot cover vast areas in real-time.



In recent years seismology has gained attention for being able to provide both temporal and spatial resolution for the detection and characterization or even forecasting of various kinds of mass movements. This includes landslides (Helmstetter and Garambois, 2010; Feng, 2011; Moore et al., 2017), rockfalls (Hibert et al., 2011; Dammeier et al., 2016; Manconi et al., 2016; Gualtieri and Ekström, 2017), avalanches (Lacroix et al., 2012; van Herwijnen et al., 2016; Hammer et al., 2017), debris flows (Walter et al., 2017) or bed load transport (Schmandt et al., 2013; Burtin et al., 2016; Roth et al., 2017). Most of the studies which demonstrate the large potential of seismology for event characterization of mass movements utilize long-period seismic radiation created by catastrophic landslides (Allstadt, 2013; Ekström and Stark, 2013; Hibert et al., 2014b). Seismic broadband observations of such events allow to invert for the 3D landslide force history and time-dependent center of mass position and - in combination with topography data - enable seismologists to fully describe a mass wasting event from remote (hundreds to thousands of kilometers distance) observations. Such observations have revealed scaling laws that link seismic observables to the mass and momentum of massive landslides (Ekström and Stark, 2013), help to constrain numerical models of landslides (Moretti et al., 2012, 2015), and support observations of frictional weakening during sliding events (Lucas et al., 2014; Levy et al., 2015; Delannay et al., 2017).

Short-period seismic radiation generated by mass movements is more complex and challenging to interpret, due to complex source mechanisms, influence of topography, directional effects, and strong near-surface scattering and attenuation. Hibert et al. (2017b) report relations between the bulk momentum of catastrophic landslides and the 3–10 Hz bandpass-filtered envelopes of the respective seismic signals. At smaller scale, controlled experiments study the generation of high-frequency seismic waves by mass impact under field (Hibert et al., 2017a) or laboratory conditions (Farin et al., 2016). Only few studies try to utilize high-frequency seismic waves to detect and characterize mass movements at local or regional scales. The majority of such studies relies on seismic data acquired in close proximity to the events, e.g. for monitoring of unstable slopes (Walter et al., 2012) or avalanches (van Herwijnen and Schweizer, 2011). Thus, although such approaches are powerful at small scale they are limited in spatial coverage (Burtin et al., 2013). Hibert et al. (2014b) demonstrate a robust automatic detection and location scheme for rockfalls inside a volcanic crater on La Réunion island. Deparis et al. (2008) first documented a set of rockfalls recorded by a regional seismic network in the western Alps and Dammeier et al. (2011) document statistical relations between rockfall characteristics and seismic recordings obtained from the Swiss permanent seismic network. Recently, there have been efforts to utilize existing regional seismic networks for the detection and characterization of mass movements (Dammeier et al., 2016; Manconi et al., 2016). Such networks - which were designed for earthquake monitoring purposes - usually consist of well-installed and sensitive seismic stations, providing high-quality seismic data in real-time and thus offer promising datasets, especially for the study of rockfalls and rockslides.

Here we present a study of 21 rockfalls and rockslides that occurred in or near Austria in the years 2007 to 2017 and were well-recorded by permanent national seismic networks in the Alps. Exploring the feasibility of a country-wide real-time detection scheme for rockfalls, we focus on automatic location routines and to automatically distinguish such events from regional earthquakes.



## 2 Dataset

This work is based on seismic recordings of 21 rockfall and rockslide events which occurred in Austria and the neighboring countries Switzerland and Italy during the years 2007 – 2017 (see Figure 1 and Table 1). Out of these 21 events, 17 rockslides have been independently verified by field observations. For photographs of the individual events please follow the references

listed at the end of the manuscript. All verified events were either first recognized by an analyst during the routine national earthquake monitoring (carried out at the Central Institute for Meteorology and Geodynamics, ZAMG) and later confirmed by field observations or were first recognized in the field and later clearly associated with seismic waveforms by analysts at ZAMG.

During routine processing of the seismic events a local magnitude $M_l$ was assigned to all rockfalls and rockslides, based on

distance and maximum amplitude as read from the seismic records, just as if the events were earthquakes. The measured local magnitude ranges between 0.0 and 2.7. For all events ground truth reference coordinates are available from field observations. However, other than date and coordinates little reliable event parameters are available, since most of the events were not studied or mapped in detail on-site, or because proper documentation could not be found.

We performed internet searches for all events listed in Table 1 to obtain on-site photographs and to find information on the

volume of rock which was displaced. For almost all events we were able to retrieve the volume which was usually reported in local newspapers, based on on-site estimates by local geological surveys. Note that these values might be subject to large uncertainties and should rather be considered as an order-of-magnitude estimation. Images for all rockfalls and rockslides are provided in the supplemental online material.

We obtained continuous waveform data for all 21 events from the European Integrated Data Archive (EIDA), which hosts

data from the permanent broadband seismic stations in the Alps. For each rockfall we identified stations within a 300 km radius around the event and downloaded all available data for all three components (Z,N,E) and at the highest sampling rate available (see Fig. 1 for network geometry). All data since 2016 is provided at 100 sps sampling rate, while earlier data is partially only available at 25 sps. For events after January 1st 2016 we also used data from the temporary AlpArray broadband stations (100 sps) which covered the entire alpine region and densify the seismic network in particular in Austria (Fuchs et al., 2016; Hetenyi

et al., 2018).

We use this dataset of confirmed events to develop and test automatic detection and locating algorithms, which we describe in the following.

## 3 Automatic processing

The first step within the automatic processing chain is the identification of a rockfall event within the continuous background

signal. We cut the seismic traces to eight-minutes segments around the known origin time (180 s prior to and 300 s after origin time) to simplify the processing and to avoid potential false alarms at this stage of development. We also restrict our processing to the vertical component only. Prior to any further processing, we remove the instrument response, apply a 1–5 Hz bandpass filter, and taper and detrend the sliced data. Note that bandpass filtering is required to enhance the signal-to-noise





**Table 1.** List of rockslides studied in this manuscript. Origin times are calculated from the seismic records. The coordinates denote the true location of the events obtained from field observations. The stations column denotes the number of stations that show a positive STA/LTA trigger. Slide volumes were obtained from a web search and are usually based on local newspaper reports – please refer to the acknowledgements section for source references. Local magnitude $M_l$ as estimated by the Austrian seismological service (ZAMG). [a] for STA/LTA threshold of 4.0 (see Section 3); [b] Not independently verified, no reference coordinates available; [c] the total volume of all three events was estimated $> 250,000$ m$^3$.

| Date | Time (UTC) | Name/Town, Country | Latitude | Longitude | Stations [a] | Volume / m$^3$ | $M_l$ |
|---|---|---|---|---|---|---|---|
| 2007-10-12 | 07:39:24 | Einserkofel, IT | 46.6390 | 12.3483 | 9 | 60,000 [1] | 2.0 |
| 2011-05-06 | 05:22:10 | Kalkkögel, AT | 47.1494 | 11.2736 | 5 | 1,000 [2] | 0.9 |
| 2011-10-23 | 14:44:34 | Tscheppaschlucht, AT | 46.4995 | 14.2769 | 12 | 30,000 [3] | 0.7 |
| 2011-12-27 | 17:25:43 | Piz Cengalo, CH | 46.2950 | 9.6020 | 74 | 1–2 $\times 10^6$ [4,5] | 2.7 |
| 2012-03-22 | 22:53:24 | Hochwand, AT | 47.3535 | 11.0041 | 24 | 150,000 [6] | 1.4 |
| 2012-05-01 | 18:26:46 | Gamsgrube, AT | 47.1179 | 11.7992 | 15 | 1–10 $\times 10^3$ [7] | 1.4 |
| 2012-05-15 | 02:45:38 | Preonzo, CH | 46.2516 | 8.9846 | 56 | 210,000 [8] | 2.2 |
| 2012-05-29 | 06:00:30 | Taschachtal, AT | 46.9186 | 10.8198 | 4 | 150,000 [9] | 0.0 |
| 2012-11-25 | 11:29:04 | Regitzer Spitz, CH | 47.0405 | 9.5012 | 6 | 180 [10] | 1.0 |
| 2014-07-13 [b] | 09:34:21 | Lienzer Dolomiten, AT | - | - | 6 | - | 0.4 |
| 2014-11-24 [b] | 16:27:20 | Trins, AT | - | - | 18 | - | 1.5 |
| 2014-11-25 [b] | 02:48:39 | Neustift im Stubaital, AT | - | - | 4 | - | 0.7 |
| 2015-01-16 | 19:22:50 | Dobratsch, AT | 46.5914 | 13.7326 | 6 | 6,000 [11] | 1.0 |
| 2015-09-30 | 20:38:18 | Schwaz, AT | 47.3485 | 1.7427 | - | 500 [12] | 0.0 |
| 2015-10-02 | 15:58:56 | Sölden, AT | 47.0051 | 10.9728 | 5 | 1–2 $\times 10^5$ [13] | 1.2 |
| 2016-03-25 | 17:14:03 | Mellental, AT | 47.3480 | 9.8400 | 45 | $> 250,000$ [c] [14] | 1.9 |
| 2016-03-25 | 21:09:02 | Mellental, AT | 47.3480 | 9.8400 | 44 | - [c] | 1.7 |
| 2016-03-26 | 01:53:00 | Mellental, AT | 47.3480 | 9.8400 | - | - [c] | - |
| 2016-05-25 | 12:51:15 | Gesäuse, AT | 47.5671 | 14.6203 | 6 | 18,000 [15] | 1.1 |
| 2016-08-19 | 21:57:04 | Kleine Gaisl, IT | 46.6425 | 12.1388 | 46 | 6–7 $\times 10^5$ [17] | 1.8 |
| 2017-02-21 | 09:36:35 | Zwölferkofel, IT | 46.6149 | 12.3749 | 40 | - | - |

ratio, especially to suppress microseism and long-period noise. Indeed, several earlier studies report this frequency band as dominant for regional seismic records of gravitational mass movements (Deparis et al., 2008; Dammeier et al., 2011; Manconi et al., 2016). Since many of the older waveform data are only available at 25 sps sampling rate, we cannot reasonably extend the bandpass window to higher frequencies. For consistency we use the same settings even for 100 sps data.





**Event detection**

For simplicity we first implemented a recursive STA/LTA coincidence trigger to detect the rockfall signals. We used the following parameters for event detection: STA window = 5 s, LTA window = 120 s, trigger-on threshold ratio = 4.0, trigger-off ratio = 1.5, minimum number of stations = 4. All events in our dataset created seismic waves strong enough to be in principle

detected with the values stated above. Table 1 lists the number of stations with positive STA/LTA trigger for each rockfall. The number of stations used for single event analysis in this study ranges from the minimum of four stations to more than 70 stations. The activation time of the STA/LTA trigger also serves as initial signal onset time for further processing.

**Kurtosis onset picker**

Once our algorithm identified stations with detectable seismic rockfall signal via the STA/LTA coincidence trigger it automati-

cally determines the signal onset on each station. Unlike earthquakes, rockfalls and rockslides commonly show rather emergent signal onsets and hence we cannot use the STA/LTA trigger times as event starting times, because the trigger-on threshold is always reached after the signal onset. Since Hibert et al. (2014a) successfully demonstrated the applicability to rockfall signals, we implemented a kurtosis-based phase picker to determine the onset of the emergent rockfall signals. The kurtosis is a statistical value, in this case characterizing the shape of a given amplitude distribution. It is a positive scalar defined as the

standardized fourth moment about the mean. In discrete form it is written as

$$\beta = \frac{\frac{1}{n}\sum_{i=1}^{n+1}(x_i - \bar{x})^4}{\left(\frac{1}{n}\sum_{i=1}^{n+1}(x_i - \bar{x})^2\right)^2} \tag{1}$$

where $n$ is the total number of samples, $x_i$ the value of the $i$-th sample, and $\bar{x}$ the mean over $n$ samples. The kurtosis of a normal distribution is $\beta = 3$ and any deviations from this value (i.e. excess kurtosis) can be used for the detection of potential seismic signals on top of regular background noise.

Similar to the processing described in Baillard et al. (2014) and Hibert et al. (2014a), we calculate a characteristic function $CF(t)$ of the seismic signal $s(t)$ within a sliding window of size $\Delta T$:

$$CF(t) = \beta\left[s(t - \Delta T), \ldots, s(t)\right] \tag{2}$$

The time window is set to $\Delta T = 5s$ and $t$ is sliding between 10s before and 1s after the preliminary onset time determined by the STA/LTA trigger. $CF(t)$ has a maximum near the true signal onset, when the kurtosis $\beta$ of the seismic amplitude

distribution within the sliding window $\Delta T$ is maximized; that is when the entire time window is dominated by seismic signals from the event (see Fig. 2). However, for location purposes we are interested in the very first onset of the seismic signal, which





is the first time $t$ at which the characteristic function $CF(t)$ starts to deviate from the background level. Thus, we adopt the procedure of Hibert et al. (2014a) and modify $CF(t)$ as follows:

$$cCF(k) = \sum_{i=1}^{k} \alpha_i \text{ with } \begin{cases} \alpha_i = CF_{i+1} - CF_i & \text{if } (CF_{i+1} - CF_i) \geq 0 \\ \alpha_i = 0 & \text{otherwise} \end{cases} \tag{3}$$

The function $cCF$ can be read as the cumulative sum of the slope of $CF$, and its value increases most drastically at the time
of the signal onset. Thus, we define the time $t$ at which the time derivative $\mathrm{d}(cCF)/\mathrm{d}t$ is maximized as the final signal onset time. If several maxima of $\mathrm{d}(cCF)/\mathrm{d}t$ lie close to each other we define the first one as the signal onset time (see Fig. 2).

**Origin time & event location**

Figure 3 shows seismic record sections for two large-scale rockslides in different areas of the eastern Alps, that show patterns of distinct seismic phase arrivals, which are common for most of the rockslides in this study. Despite the emergent character
of the rockslide signals we can identify a first arrival that travels with an apparent velocity of approximately 5.0 km/s. We thus assume that this arrival is a P wave. For some events a distinct second arrival is visible, which is usually stronger than the first arrival and sometimes (in case of low signal-to-noise ratio) is the only visible signal. This arrival travels with an apparent velocity of approximately 3.0 km/s and we suggest that it is due to S waves or surface waves (see Discussion section).

We run a grid search to estimate the origin time and location of the rockslides based on the onset times determined by the
kurtosis picker. The search area is a rectangle with 5 km grid spacing spanned by all seismic stations with positive STA/LTA detection. Time is scanned in steps of 2s between the earliest measured onset time (= latest possible origin time) and an estimated earliest possible origin time which is set as the first onset pick minus the maximum travel time along the grid diagonal. For each grid point and each time step we calculate the theoretical arrival time (fixed velocity of 5.0 km/s, no topography) for all stations and its difference (= residual) to the measured onset time. The grid point and time where the root-
mean-square (RMS) value of the set of station residuals is minimized is set as preliminary origin time and event location (see Fig. 4). For one third of the rockslides analyzed within this study the simple grid search location is already quite satisfactory, with results that are significantly less than 10 km from the true rockslide location.

To overcome the simplifications of the grid search we subsequently perform an iterative location routine as is done for earthquakes, using the HYPOCENTER code (Havskov and Ottemoller, 1999) and a simple 1D velocity model suitable for
the eastern Alps (Hausmann et al., 2010). The kurtosis-based onset picks are treated as crustal Pg waves. The results are summarized in Table 2 and demonstrate the location accuracy which can be achieved even for emergent rockslide signals with regional seismic records. 8 of 18 tested events were located less than 6 km from the true location. 4 events could not be located due to very low signal-to-noise ratio or insufficient number of stations. We discuss possible limitations and reasons for outliers as well as the robustness of the results in the discussion section below.



**Table 2.** Location quality based on kurtosis picks. The deviation indicates the discrepancy between the final location result and the true location of the event. Four events could not be located due to insufficient number of picks. [a] Number of stations (= number of picks) used for location routine; this number may deviate from the number of stations that passed the STA/LTA trigger (see Table 1) because the kurtosis algorithm may not have found viable onset picks. [b] Only the strongest event from the sequence is listed.

| Date | Time (UTC) | Name/Town, Country | Stations [a] | Azimuthal Gap / ° | Deviation / km |
|---|---|---|---|---|---|
| 2012-05-15 | 02:45:38 | Preonzo, CH | 56 | 54 | 0.7 |
| 2015-01-16 | 19:22:50 | Dobratsch, AT | 5 | 273 | 3.7 |
| 2015-10-02 | 15:58:56 | Sölden, AT | 5 | 183 | 4.3 |
| 2016-08-19 | 21:57:04 | Kleine Gaisl, IT | 44 | 41 | 4.3 |
| 2012-05-01 | 18:26:46 | Gamsgrube, AT | 12 | 147 | 4.8 |
| 2016-03-25 | 17:14:03 | Mellental, AT [b] | 40 | 64 | 5.0 |
| 2011-10-23 | 14:44:34 | Tscheppaschlucht, AT | 9 | 153 | 5.6 |
| 2012-11-25 | 11:29:04 | Regitzer Spitz, CH | 4 | 141 | 5.8 |
| 2011-12-27 | 17:25:43 | Piz Cengalo, CH | 73 | 87 | 8.3 |
| 2012-03-22 | 22:53:24 | Hochwand, AT | 27 | 175 | 8.3 |
| 2007-10-12 | 07:39:24 | Einserkofel, IT | 9 | 145 | 8.8 |
| 2011-05-06 | 05:22:10 | Kalkkögel, AT | 4 | 187 | 11 |
| 2016-05-25 | 12:51:15 | Gesäuse, AT | 5 | 206 | 16 |
| 2014-11-24 | 16:27:20 | Trins, AT | 18 | 134 | 26 |
| 2012-05-29 | 06:00:30 | Taschachtal, AT | - | - | - |
| 2014-07-13 | 09:34:21 | Lienzer Dolomiten, AT | - | - | - |
| 2014-11-25 | 02:48:39 | Neustift im Stubaital, AT | - | - | - |
| 2015-09-30 | 20:38:18 | Schwaz, AT | - | - | - |

**Discrimination from regional earthquakes**

A key aspect for automatic processing of seismic rockslide data is to distinguish such events from earthquakes and other potential sources of seismicity. Hibert et al. (2014a) suggest a set of parameters that are extracted from the seismic signal and are systematically different for earthquakes and rockslides. Here we explore if this simple concept that was successfully applied on local scale can be extended to regional scale.

For each rockslide signal on each available station we extract the following information (see Fig. 5): the Kurtosis of the envelope of the entire signal (*EnvKurto*); the ratio between maximum amplitude and mean amplitude (*Max/Mean*); the ratio of the duration (*Inc/Dec*) of the increasing signal flank (signal start to maximum amplitude) compared to the duration of the decreasing signal flank (maximum amplitude to signal end). The end time of the event is defined as the time where the 2s moving average of the signal envelope decayed to $1.1 \times$ the pre-event levels. The pre-event amplitude is estimated as the mean amplitude within a 60s window 5s prior to the first signal onset.



We extract the same parameters from a set of regional earthquake records in order to identify potential differences between rockslides and earthquakes. We downloaded data for 32 earthquakes ($M_l < 3.5$) within 08/2015 and 01/2016 that occurred in or near western Austria. Thus, the earthquakes occurred in the same area as the rockslides and induced similar levels of shaking. The processing of the earthquake data was the same as for the rockslide data and we read the parameters described above for

each earthquake on each available station.

Figure 5 shows the distribution of potential discrimination parameters extracted from rockslides and earthquakes. For all parameters both distributions overlap but they peak at different values. Notably, for rockslides all values measured for the kurtosis of the envelope (*EnvKurto*) and the ratio of maximum-to-mean amplitude (*Max/Mean*) stay below a certain threshold, as compared to earthquakes. We make use of this observation and define a simple decision criterion whether an event should be

declared as rockslide or earthquake. An event is considered as a rockslide if the mean value measured over all stations satisfies the following condition:

$$\log(\textit{EnvKurto}) < 1.2 \quad \text{AND} \quad \log(\textit{Max/Mean}) < 1.2 \quad \text{AND} \quad \log(\textit{Inc/Dec}) > -1.1 \tag{4}$$

This way all 21 rockslides and all 32 regional earthquakes are correctly identified and we demonstrate that even on regional scale it might be possible to distinguish rockslides from earthquakes based on a few simple criteria. We introduce potential

extensions of this scheme in the discussion section.

**Volume-Magnitude relation**

Beside the event location the event volume is a crucial parameter for an assessment of a rockslide. Thus we attempt to relate the slide volume to the local magnitude $M_l$, a parameter that is routinely determined for seismic events by any seismological service. Several studies (Deparis et al., 2008; Dammeier et al., 2011; Ekström and Stark, 2013; Hibert et al., 2014a) attempt

to relate the volume of mass movements to the measured seismic energy or amplitude. However, derived scaling relations are often only loosely constrained due to e.g. limited number of events, generally large scatter or insufficient information about the event. From the 21 events studied here, 15 rockslides that have a well-defined magnitude and a volume estimate available (see Table 1). Figure 6 shows the local Magnitude as a function of the event volume. Although the proposed fit is not very well constrained ($R^2 = 0.35$) due to large scatter and limited data points, the distribution suggests a linear relation between the local

magnitude $M_l$ and the logarithmic volume $V$:

$$M_l = -0.43 + 0.38 \log V \tag{5}$$



Since the local magnitude $M_l = \log(A/A_0)$ is a logarithmic measure of the seismic amplitude $A$ this translates into a power law relation between the seismic amplitude $A$ and the rockslide volume $V$, including a regional correction term $A_0$ which depends on the epicentral distance corrections applied during the calculation of $M_l$:

$$A = A_0 \left(0.37 + V^{0.38}\right) \qquad (6)$$

## 4  Discussion

Here we demonstrated that regional seismic networks can be used to reliably detect moderate to large-size rockslides to distances up to more than 200 kilometers. Such seismic networks cover vast areas and record data continuously, and many networks provide data in real-time. Thus, they allow for regional monitoring of potentially catastrophic mass movements, and they additionally provide a temporal resolution which is unmatched by classical methods such as remote sensing. Here we suggest several processing steps to analyze the seismic signal generated by rockslides and show that simple concepts and easy-to-integrate tools already provide reasonable insight into the events. This demonstrates that even large datasets may be screened for rockslide data automatically. While this shows the potential of regional seismic records to study gravitational mass movements, there is much room for improvements which may strongly increase the quality of the extractable information. All processing steps including the event location and characterization were performed completely automatically without intervention of a human analyst. In particular no attempt was made to remove outliers or e.g. wrong onset picks, which in some cases greatly reduces the quality of the location result. Still, our simplistic approach may be complemented in most of the processing steps to increase the robustness of the results.

### Event detection

We have shown that all moderate to large-size rockslides in this study could in principle be detected with a STA/LTA coincidence detector which is widely used by e.g. seismological observatories and generally serves as a fast algorithm to screen datasets for events. However, STA/LTA detectors need to be balanced between sensitivity and rate of false alarms. While the STA/LTA settings reported above do safely detect all of our events we did not check how many false alarms would be introduced if a continuous data stream was analyzed (we cut the data to eight minutes around the events). Generally, there are more sensitive yet more computationally intensive algorithms to detect events in continuous seismic data. Dammeier et al. (2016) demonstrate how alpine rockslides can be automatically detected on regional networks using Hidden Markov Models, which allows to simultaneously detect and classify mass movements within seismic records.

### Kurtosis picker performance & location accuracy

Hibert et al. (2014a) designed a robust onset picker for rockslide signals based on a transition in the kurtosis. However, the method was only applied at very local scale (network extension of few kilometers) around a volcano. Baillard et al. (2014)





also document the performance of a kurtosis picker for earthquake localization on regional seismic networks. Here we show that this concept could also be applied to the rather emergent signals induced by gravitational mass movements. Eight of 14 locatable events in this study could be located within few kilometers deviation from the true location (see Table 1), which shows that based on onset picks a similar precision as for earthquakes is possible. However, some of the locations should be considered *lucky hits*, as e.g. the number of stations is low and the azimuthal gap is large, strikingly for some of the most well-located events. We do in fact observe that the location results currently lack robustness and may change by few kilometers when certain parameters of the kurtosis picker (e.g. the length of the moving window) are adjusted. This is most likely due to both unfavorable noise conditions and to the simplistic processing which we used for demonstration purposes. We expect that picking accuracy can be greatly improved if measures are taken to make the kurtosis picker more robust. Future work should include all three components of the seismic record and use different frequency bands for comparison, as suggested by Hibert et al. (2014a). We expect that such measures would suppress outliers due to e.g. noise and thus make the onset determination more robust and precise. In this study - due to limited sampling rate for older records - we were limited to the 1–5 Hz range. Note however, that many studies report this frequency range as the dominant one for regional seismic records of rockslides (Deparis et al., 2008; Dammeier et al., 2011; Manconi et al., 2016).

Besides kurtosis methods, pickers based on e.g. autoregressive prediction (Küperkoch et al., 2012) might be very suitable for emergent onset picks, as they include frequency and phase information in addition to the amplitude (kurtosis pickers are only based on amplitudes). Since determining the onset of an emergent signal is anyways challenging, pickless location routines such as waveform correlation (Arrowsmith et al., 2016) should also be explored for mass movements. Manconi et al. (2016) suggest to combine location probabilities obtained from seismic waves with location probabilities based on terrain slopes to narrow down the potential source areas.

For location purposes we assumed the first onset of the rockslide signals to be a direct i.e. crustal P-wave. The observed average phase velocity of the first arrival is approximately 5.0 km/s (see Fig. 3), which is similar to the observations by Dammeier et al. (2011) and represents a typical value for P-wave velocities in the upper crust of the Eastern Alps (Ye et al., 1995; Husen et al., 2003; Hausmann et al., 2010). For some events a very distinct second arrival is visible (see Fig. 3b) that travels at lower velocities of approximately 3.0 km/s. In this velocity range we potentially expect both crustal S-waves or surface waves. If the type of wave was clearly identifiable a second phase pick would be available which could drastically increase the location accuracy. Other events (Fig. 3a) show no clear second onset and amplitudes gradually increase towards the maximum after the first onset. This *cigar-type* shape is more commonly found in other seismic studies of landslides and rockslides (Deparis et al., 2008; Dammeier et al., 2011; Hibert et al., 2014a). For such events we observe that the signal group around the maximum amplitude travels slower than the first onset, which suggests that P-waves and other type of waves mix within the signal and complicate any in-detail analysis of the seismic phases or polarization. The mechanism of each individual rockslide event likely influences the relative strength at which certain wave types are generated. We also suggest that depending on the slide mechanism e.g. P-waves and S-waves must not necessarily be excited at the same time during the event. Additionally, a rockslide potentially is a very directional source of seismic energy which may introduce anisotropic radiation patterns for the seismic energy. Wang et al. (2016) point out the influence of scattering at surface topography for



location purposes and we should note that gravitational mass movements might be particularly affected by such effects since they occur in areas of pronounced topography and at the earth surface.

**Event discrimination & volume estimation**

We demonstrate that rockslides and earthquakes from the same source region can be discriminated by few simple parameters
such as the ratio between maximum and mean amplitude of the seismic signal or the amplitude distribution. Manconi et al. (2016) present a robust decision criterion only based on the ratio $M_l/M_d$ of the local magnitude $M_l$ and the duration magnitude $M_d$. Hibert et al. (2014a) proposed to combine several criteria within a simple fuzzy-logic decision algorithm and we suggest that similar approaches can safely distinguish rockslides from earthquakes also on regional scale. Note however, that each region where such methods are applied might require individual modification of the decision thresholds for each parameter.
Recently, more sophisticated decision algorithms based on machine learning have been developed that allow to classify any kind of seismic event within a huge event database with great precision, after being trained by a few selected known events. Classifiers based on random forest algorithms were successfully applied to classify gravitational mass movements and other events in several different settings, such as volcanoes (Maggi et al., 2017) or slow-moving landslides (Provost et al., 2017) and show great potential for the application on regional seismic networks (Hibert et al., 2018).

Extracting reliable volume or mass information from the seismic records of mass movement remains challenging and requires more research on the factors influencing the efficiency of seismic wave generation. Among these factors are e.g. the bulk mass, the drop mechanisms (free fall and impact versus sliding), the slope and the runout distance. For catastrophic events that generate strong long-period signals, such properties can be inverted for from the seismic data (Allstadt, 2013; Ekström and Stark, 2013; Hibert et al., 2014b). Short-period radiation is more complex to interpret though. Hibert et al. (2017b) report
simple scaling relations between the bulk mass momentum and short-period seismic amplitudes for catastrophic landslides from within the same source area, if source mechanisms are comparable among different events. They report similar observations also for controlled single-block fall experiments (Hibert et al., 2017a). At local scale, knowledge of the topography and a large number of events helps to constrain parameter estimates based on the seismic signals (Hibert et al., 2014a). At regional scale however, unknown scattering, attenuation, and propagation of the short period seismic waves may obscure any potential
scaling relations.

Deparis et al. (2008) point out that regional attenuation relations extracted from earthquakes may not be applicable to rockfall records and thus local magnitudes may not properly reflect the amount of seismic energy released by the source. They suggest that peak ground velocity is not a good measure to characterize rockfall signals. In contrast, Dammeier et al. (2011) deduct reasonably well-constrained relationships between rockslide parameters and the seismic peak ground velocity.
This is in agreement with our findings that show an acceptable power-law relation between the averaged maximum seismic amplitude and the slide volume. Dammeier et al. (2011) suggest that regional propagation and attenuation of rockslide signals is strongly influenced by topography. In addition, several studies observe that the seismic efficiency - the ratio of available potential energy over the released seismic energy - is usually low for gravitational mass movements (Deparis et al., 2008; Ekström and Stark, 2013; Hibert et al., 2014a). This may in part explain the poor correlations between seismic amplitudes





and the rockslide volumes for several studies (including this one), since it suggests that a large part of the potential energy is released through other processes (e.g. friction, cracking, plastic deformation) and not transmitted seismically (Deparis et al., 2008). Manconi et al. (2016) attempt to derive a scaling law for the rockslide volume not based on seismic amplitudes but on the duration magnitude $M_d$ and they show a reasonable empirical correlation even for events of very different mechanisms and origin areas.

A general drawback of many studies that aim to identify scaling relations for seismic energy created by gravitational mass movements at regional scale is the limited number of events (Deparis et al., 2008; Dammeier et al., 2011; Manconi et al., 2016). This is partly due to the limited availability of high-quality seismic data (network density, sampling rate), geographical restrictions (e.g. country borders) or lack of reliable event information (e.g. volume). Advancing our knowledge about short-period seismic radiation created by gravitational mass movements now calls for several actions: Merging or cross-checking of national event databases - which unfortunately often end at country borders - should greatly improve the number of events available for analysis and the robustness of the event parameters. Multidisciplinary approaches should be explored to constrain event parameters routinely also via e.g. remote sensing. Finally, efficient data screening algorithms will allow to detect and classify gravitational mass movements inside huge datasets, such as the AlpArray seismic network (Hetenyi et al., 2018). This will drastically increase the number of events to study and thus opens new possibilities to investigate the triggers of and mechanisms during gravitational mass movements.

## 5  Conclusion

We have demonstrated how to search for seismic signatures of rockslides in the data of regional seismic networks up to more than 200 km from the origin. Kurtosis-based phase pickers allow to reliably detect the onset of rockslide signals despite their emergent character. Resulting location accuracies are in the range of a few kilometers and can potentially be greatly reduced by incorporating proper handling of outliers and if secondary phases can be clearly associated. Automatic discrimination from earthquakes and other local or regional sources is possible by a simple combination of three decision parameters, such as maximum-to-mean amplitude ratio. Based on a larger set of similar parameters, future application of machine learning techniques to the data of regional seismic networks promises automatic event classification with great accuracy. This will likely increase the number of seismically detected rockslide events at regional scale. Larger and better parameterized data sets of rockslides will clarify scaling relations between event parameters and seismic observables, and will help to better understand the seismic waves created by gravitational mass movements. Regional seismic networks can cover vast areas and at the same time provide continuous data for very long time series. This combination of spatial coverage and temporal resolution is currently unmatched by other geophysical methods. Thus, seismic networks are ideally suited to remotely study time-dependent rockslide activity. This may include e.g. long-term variations in rockslide activity potentially linked to climate change, fore- and afterslides of a main event, and a more detailed insight into rockslide triggering factors



## Data availability and methods

The majority of seismic waveform data used in this study is openly available for download at the European Integrated Data Archive (EIDA, http://www.orfeus-eu.org/data/eida/index.html, last accessed June 2018). Waveform data with network code Z3 was acquired from the temporary AlpArray Seismic Network (2015), which at the time of publication was not openly

available by decision of the AlpArray Working Group. Please visit www.alparray.ethz.ch (last accessed June 2018) for a complete description of data access.

All processing required for this manuscript was done using the ObsPy toolbox (Krischer et al., 2015; The ObsPy Development Team, 2017). For location purposes we made use of certain modules of the Seisan analysis software package (Havskov and Ottemoller, 1999).

Rockslide photographs and references for volume estimations in Table 1:

[1] http://tirv1.orf.at/stories/228199

[2] http://tirv1.orf.at/stories/514304

[3] http://kaernten.orf.at/news/stories/2506673

[4] www.srf.ch/play/tv/news-clip/video/fast-unbemerkt-riesen-bergsturz-im-bergell?id=6f9ce66d-6c9b-47c3-9842-5ee19531af57

[5] http://www.zeit.de/2014/36/bergell-bergsturz-schweiz

[6] Geoforum Tirol, Tagungsband, 14. Geoforum Umhausen, 2012

[7] https://www.meinbezirk.at/telfs/lokales/heuer-bereits-vier-mal-soviele-einsaetze-wie-im-vergleich-zum-vorjahr-d212155.html

[8] Loew et al. (2017) (see below)

[9] http://tirol.orf.at/news/stories/2535035

[10] http://www.vilan24.ch/Flaesch.114.0.html?&cHash=0a607912512d9efae1fe768fb2a36494&tx_ttnews%5Btt_news%5D=7719

[11] https://www.zamg.ac.at/cms/de/geophysik/news/massiver-felssturz-am-dobratsch-bei-villach

[12] https://www.tirol.gv.at/meldungen/meldung/artikel/ersteinschaetzung-der-landesgeologie-keine-gefahr-fuer-siedlungsraum

[13] http://www.tt.com/panorama/natur/10657382-91/%C3%B6tztaler-felssturz-kam-einem-erdbeben-gleich.csp

[14] E. Vigl, Aktenvermerk VIIa-68.010.58-1//-222, Amt der Vorarlberger Landesregierung, Bregenz, 18/04/2016

[15] J. Reinmüller, https://host14.ssl-net.net/xeis-auslese_at/wp-content/uploads/2016/05/Dachl-Felssturz.pdf

[16] http://www.tt.com/panorama/natur/11727492-91/nach-felssturz-in-hopfgarten-land-baut-sicherheitsdamm.csp

[17] https://www.stol.it/Artikel/Chronik-im-Ueberblick/Lokal/Erwin-Steiner-Es-rumpelte-ueberall-in-Prags

## Team list

György Hetényi (University of Lausanne, Switzerland), Rafael Abreu (University of Münster, Germany), Ivo Allegretti (University of Zagreb, Croatia), Maria-Theresia Apoloner (University of Vienna, Austria), Coralie Aubert (University Grenoble Alpes, France), Simon Besançon (IPGP, France), Maxime Bés de Berc (University of Strasbourg, France), Götz Bokelmann (University of Vienna, Austria), Didier Brunel (University Nice Sophia Antipolis, France), Marco Capello (University of Genova, Italy), Martina Čarman (ARSO, Slovenia), Adriano Cavaliere (INGV, Italy), Jérôme Chéze (University Nice Sophia Antipolis, France), Claudio Chiarabba (INGV, Italy), John Clinton (SED, Switzer-

land), Glenn Cougoulat (University Grenoble Alpes, France), Wayne C. Crawford (IPGP, France), Luigia Cristiano (Christian-Albrechts-



University Kiel, Germany), Tibor Czifra (Hungarian Academy of Sciences, Hungary), Ezio D'alema (INGV, Italy), Stefania Danesi (INGV, Italy), Romuald Daniel (IPGP, France), Anke Dannowski (GEOMAR, Germany), Iva Dasović (University of Zagreb, Croatia), Anne Deschamps (University Nice Sophia Antipolis, France), Jean-Xavier Dessa (CRNS, France), Cécile Doubre (University of Strasbourg, France), Sven Egdorf (University of Munich, Germany), ETHZ-SED Electronics Lab (SED/ETH Zurich, Switzerland), Tomislav Fiket (University of

Zagreb, Croatia), Kasper Fischer (Ruhr University Bochum, Germany), Wolfgang Friederich (Ruhr University Bochum, Germany), Florian Fuchs (University of Vienna, Austria), Sigward Funke (University of Leipzig, Germany), Domenico Giardini (ETH Zurich, Switzerland), Aladino Govoni (INGV, Italy), Zoltán Gráczer (Hungarian Academy of Sciences, Hungary), Gidera Gröschl (University of Vienna, Austria), Stefan Heimers (SED, Switzerland), Ben Heit (GFZ Potsdam, Germany), Davorka Herak (University of Zagreb, Croatia), Marijan Herak (University of Zagreb, Croatia), Johann Huber (University of Vienna, Austria), Dejan Jarić (Republic Hydrometeorological Service

of Republic of Srpska, Bosnia and Herzegovina), Petr Jedlička (Czech Academy of Sciences, Czech Republic), Yan Jia (ZAMG, Austria), Hélène Jund (University of Strasbourg, France), Edi Kissling (ETH Zurich, Switzerland), Stefan Klingen (University of Münster, Germany), Bernhard Klotz (Ruhr University Bochum, Germany), Petr Kolínský (University of Vienna, Austria), Heidrun Kopp (GEOMAR, Germany), Michael Korn (University of Leipzig, Germany), Josef Kotek (Czech Academy of Sciences, Czech Republic), Lothar Kühne (Ruhr University Bochum, Germany), Krešo Kuk (University of Zagreb, Croatia), Dietrich Lange (GEOMAR, Germany), Jürgen Loos (University

of Munich, Germany), Sara Lovati (INGV, Italy), Deny Malengros (Mediterranean Institute of Oceanography, France), Lucia Margheriti (INGV, Italy), Christophe Maron (University Nice Sophia Antipolis, France), Xavier Martin (University Nice Sophia Antipolis, France), Marco Massa (INGV, Italy), Francesco Mazzarini (INGV, Italy), Thomas Meier (University of Kiel, Germany), Laurent Métral (University Grenoble Alpes, France), Irene Molinari (ETH Zurich, Switzerland), Milena Moretti (INGV, Italy), Helena Munzarová (Czech Academy of Sciences, Czech Republic), Anna Nardi (INGV, Italy), Jurij Pahor (ARSO, Slovenia), Anne Paul (University Grenoble Alpes, France),

Catherine Péquegnat (University Grenoble Alpes, France), Daniel Petersen, Damiano Pesaresi (OGS Udine, Italy), Davide Piccinini (INGV, Italy), Claudia Piromallo (INGV, Italy), Thomas Plenefisch (BGR, Germany), Jaroslava Plomerová (Czech Academy of Sciences, Czech Republic), Silvia Pondrelli (INGV, Itlay), Snježan Prevolnik (University of Zagreb, Croatia), Roman Racine (SED, Switzerland), Marc Régnier (University Nice Sophia Antipolis, France), Miriam Reiss (University of Frankfurt, Germany), Joachim Ritter (KIT, Germany), Georg Rümpker (University of Frankfurt, Germany), Simone Salimbeni (INGV, Italy), Marco Santulin (INGV, Italy), Werner Scherer (KIT, Ger-

many), Sven Schippkus (University of Vienna, Austria), Detlef Schulte-Kortnack (University of Kiel, Germany), Vesna Šipka (Republic Hydrometeorological Service of Republic of Srpska, Bosnia and Herzegovina), Stefano Solarino (INGV, Italy), Daniele Spallarossa (University of Genova, Italy), Kathrin Spieker (University of Leipzig, Germany), Josip Stipčević (University of Zagreb, Croatia), Angelo Strollo (GFZ Potsdam, Germany), Bálint Süle (Hungarian Academy of Sciences, Hungary), Gyöngyvér Szanyi (Hungarian Academy of Sciences, Hungary), Eszter Szűcs (Hungarian Academy of Sciences, Hungary), Christine Thomas (University of Münster, Germany), Martin Thor-

wart (University of Kiel, Germany), Frederik Tilmann (Free University Berlin, Germany), Stefan Ueding (University of Münster, Germany), Massimiliano Vallocchia (INGV, Italy), Luděk Vecsey (Czech Academy of Sciences, Czech Republic), René Voigt (University of Leipzig, Germany), Joachim Wassermann (University of Munich, Germany), Zoltán Wéber (Hungarian Academy of Sciences, Hungary), Christian Weidle (University of Kiel, Germany), Viktor Wesztergom (Hungarian Academy of Sciences, Hungary), Gauthier Weyland (University of Strasbourg, France), Stefan Wiemer (SED, Switzerland), Felix Wolf (GEOMAR, Germany), David Wolyniec (University Grenoble Alpes,



France), Thomas Zieke (GFZ Potsdam, Germany), Mladen Živčić (ARSO, Slovenia).

*Acknowledgements.* This work was funded by the Austrian Science Fund FWF project number P26391. This work did benefit from fruitful discussions at the EGU Galileo conference on Environmental Seismology 2017, Ohlstadt, Germany.

5    We thank Helmut Hausmann (ZAMG) for his help to compile the event parameters and independent information. Nils Tilch and Alexandra Haberler of the Geological Survey of Austria (GBA) are thanked for the cooperation and help in compiling the event database, verification of seismic data and alerting us of new rockslides.

We acknowledge the use of data from the AlpArray network (code Z3; AlpArray Seismic Network (2015)) - please visit the project homepage www.alparray.ethz.ch for a full list of people contributing to the AlpArray seismic network.

10    For this study we used seismic data from several permanent seismic networks and we appreciate the continuous operation of these seismic networks by the responsible institutions: BW net (Department of Earth and Environmental Sciences, Geophysical Observatory, University of Munchen, 2001), CH net (Swiss Seismological Service (SED) at ETH Zurich, 1983), CR net, FR net (RESIF, 1995), GN net (Institut de Physique du Globe de Paris (IPGP) & Ecole et Observatoire des Sciences de la Terre de Strasbourg (EOST), 1982), GU net (University of Genova, 1967), GR net, IV net (INGV Seismological Data Centre, 1997), MN net (MedNet project partner institutions, 1988), NI net

15    (OGS (Istituto Nazionale di Oceanografia e di Geofisica Sperimentale) and University of Trieste, 2002), OE net, OX net (OGS (Istituto Nazionale di Oceanografia e di Geofisica Sperimentale), 2016), SI net, SL net (Slovenian Environment Agency, 2001), and ST net (Geological Survey-Provincia Autonoma di Trento, 1981). We acknowledge ORFEUS and EIDA for providing the tools to access the seismic data.





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





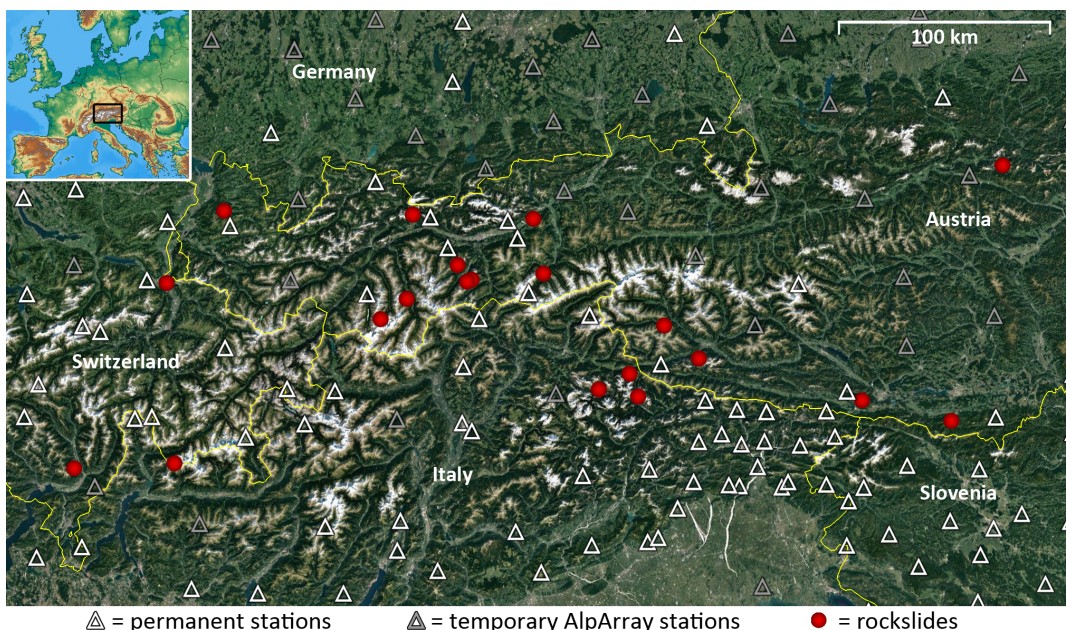

**Figure 1.** Map of the study area in eastern Austria and neighboring countires. Rockslides are marked by red circles. Bright and dark triangles denote permanent and temporary seismic stations, respectively. The yellow lines mark country borders. The inset marks the location of the study area in Europe.



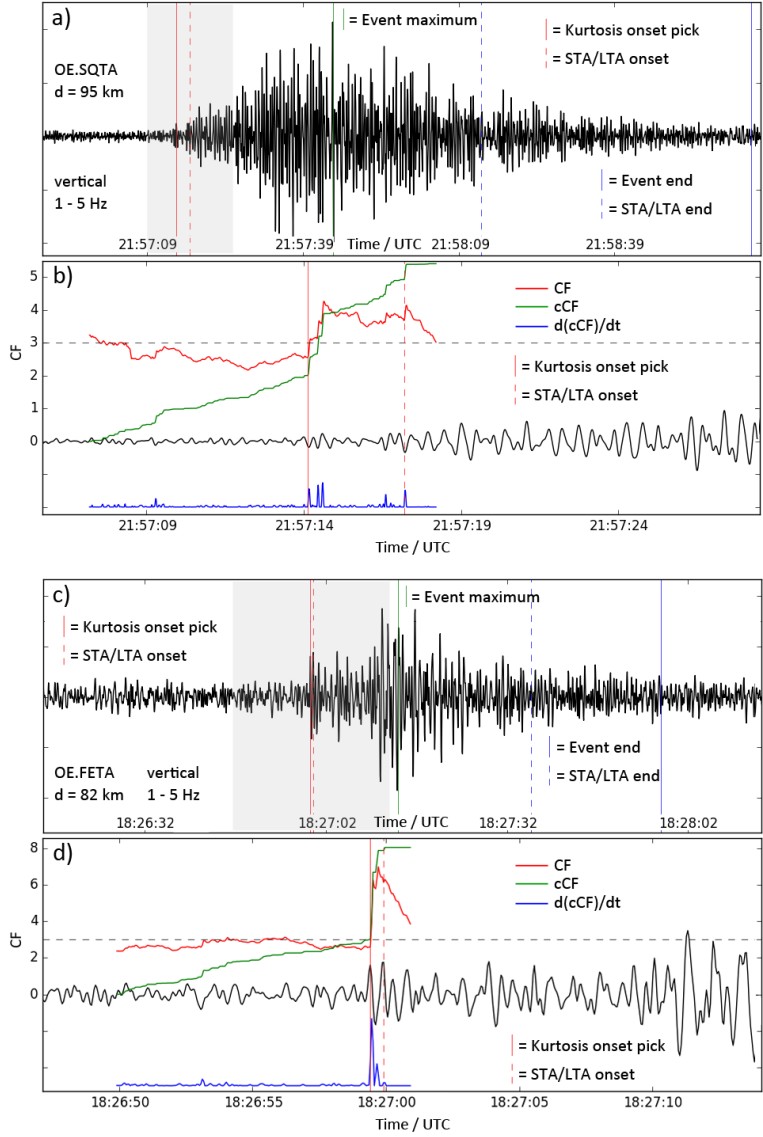

**Figure 2.** Examples for performance of the kurtosis picker. All waveforms are from 1-5 Hz bandpass-filtered vertical components. The upper panels (a,b) show an example of the 2016-08-19, Kleine Gaisl, Italy rockslide from station OE.SQTA at 95 km distance. The bottom panels (c,d) show an example of the 2012-05-01, Gamsgrube, Austria rockslide from station OE.FETA at 82 km distance. Panels b) and d) show close-ups of the grey-shaded parts of the waveforms in panels a) and c), respectively. The vertical axes in panels b) and d) indicate the values of $CF$. For perfectly-gaussian noise we expect a value $CF = 3.0$, which is marked by the dashed horizontal lines. Vertical lines denote picks for the event onset and end. Solid red line: onset pick based on maximum $d(cCF)/dt$. Dashed red line: onset time of STA/LTA trigger. Solid blue line: Event end time as given by the $1.1 \times$ pre-event noise condition (see Section 3). Dashed blue line: End time of STA/LTA trigger (for comparison; not used for any processing).





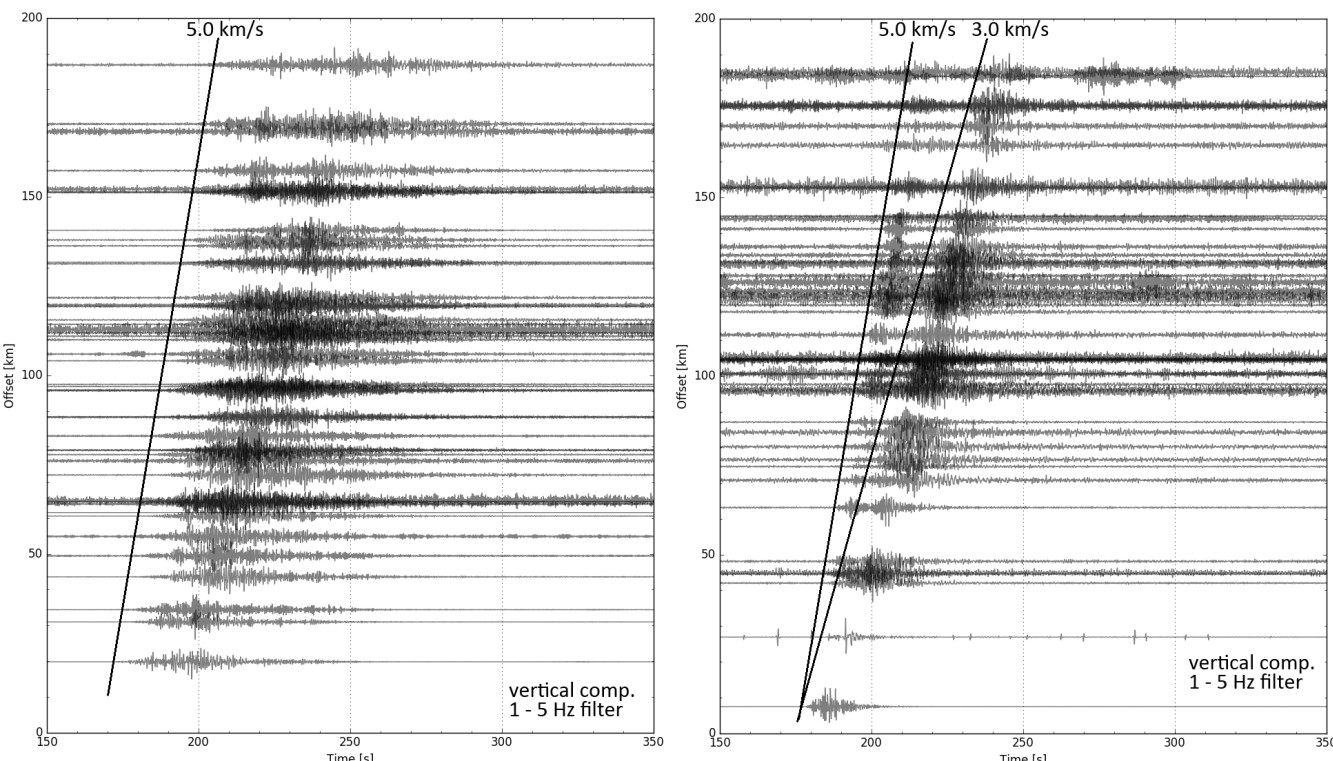

**Figure 3.** Record sections (signal vs. distance) of the vertical component for two large rockslides. All data are bandpass-filtered between 1 and 5 Hz. Left: Kleine Gaisl, Italy, 2016-08-19, as an event example that does not show a clear second arrival. Right: Mellental, Austria, 2016-03-25, which does show a distinct second arrival for stations farther than 50 km from the origin. Black lines mark expected arrival times for a constant travel-time of 5.0 km/s and 3.0 km/s, respectively.




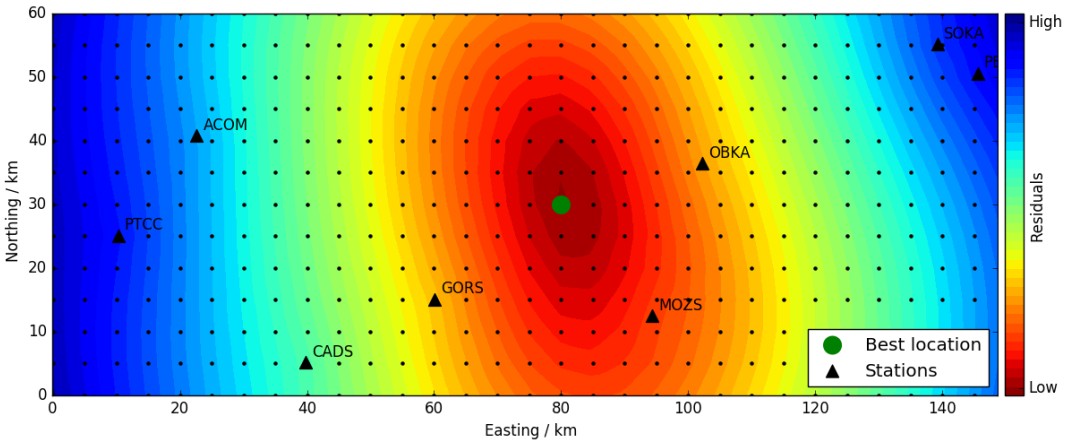

**Figure 4.** Example for a grid-search result (rockfall in Tscheppaschlucht, Austria, 2011/10/23). Black triangles mark the stations used for the grid search. Colors indicate the root-mean-square travel time residuals among all stations (for the best fitting origin time and for a fixed velocity of 5.0 km/s). Note that colors are smoothed between grid points (small black dots). The green dot represents the grid point that minimizes the set of travel time residuals and thus marks the preliminary location of the rockslide.

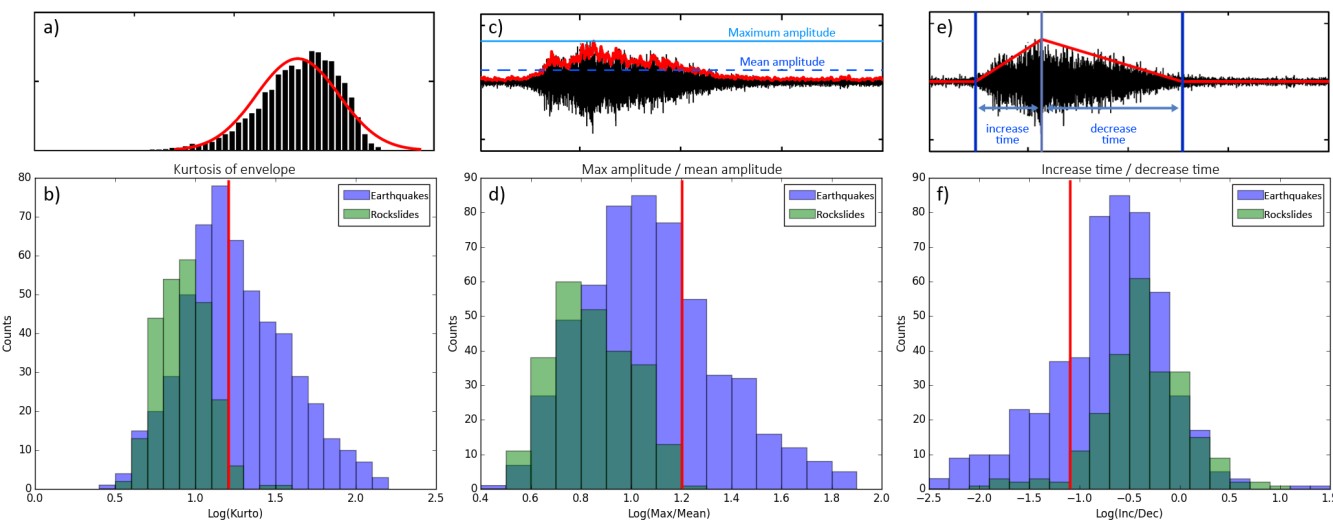

**Figure 5.** Distributions of different discrimination parameters for rockslides and earthquakes. Upper panels (a,c,e) show the definition of the respective parameters. Lower panels (b,d,f) show the frequentness of the respective parameters in logarithmic scale. Note that the total number of parameter reads is slightly higher for earthquakes than for rockslides and the distributions are not normalized. Green colors marks the values read from rockslide records, blue colors mark the values read from earthquake records. The red lines in panels b,d,f mark the respective thresholds for the decision criterion (see Eq. 4).



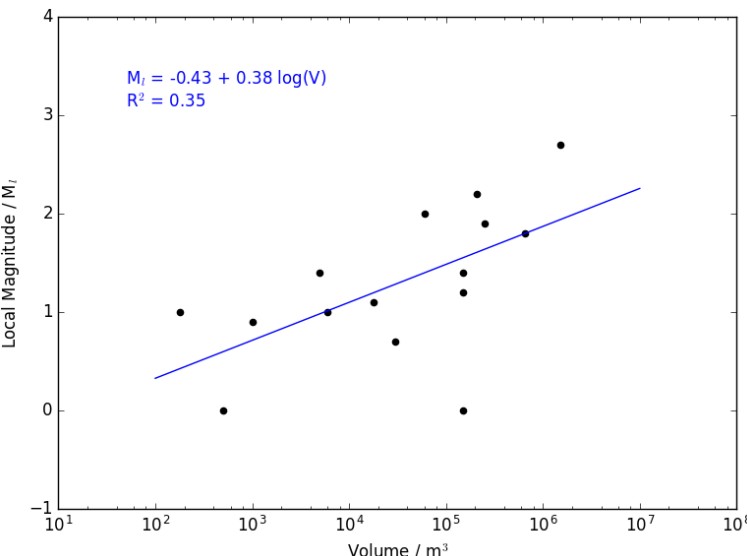

**Figure 6.** Local magnitude of all rockslides versus their volume (black dots). The distribution indicates a linear relation (blue line) between magnitude and logarithmic volume. The equation with the best-fitting parameters and the coefficient of determination $R^2$ are indicated above the graph.