# Peer review of "Seismic detection of rockslides at regional scale: Examples from the Eastern Alps and feasibility of kurtosis-based event location"

_Earth Surface Dynamics, 2018_

## Referee Comment (RC1) · Anonymous Referee #1 · 30 Jul 2018

In this paper, the authors use the seismic data to automatically detect the rockslide and locate the events in the Eastern Alps. It's an important work for the practical application in the future. However, there are some major issues need to be classified.

1. The authors mentioned that the purpose of research focuses on automatic location and automatically distinguish earthquake events. But the research uses the known database and construct the algorithms. I would suggest the authors revise the purpose of this research.

2. From the reference photo, some rock disaster seems like a free fall event. It has better to define rockslide in the introduction part.

[Figure]

3. In page 5, The STA/LTA method common uses in signal analysis. I think the authors should add some references in this part. Further, why the authors set the different thresholds of trigger-on and trigger-off ratio?

4. In page 6, Many research support that the P and S wave can't be classified in rockslide/landslide events. From the right part of the Fig.3, the event also looks like containing two parts. The minor event happened first and a major event occurred following. It's a common situation in rockslide event. I suggest the authors carefully check the data again. If this is P and S wave, I think the authors should describe it in detail.

5. From the automatic detection, I think it may detect some unknown rockslides, but from authors' data, all rockslides are known events. In advance, I would suggest the authors use different values (like different frequency range) to run the automatic detection. And some deviations of location are quite large. It's a little bit impractical for further application.

6. The authors construct eq.(4) to distinguish the earthquakes and rockslides. I would suggest the authors validate the equation with rockslides after February 2017.

7. From the fig 6, two events' local magnitude is zero. If the authors remove these two points, the R2 should be higher. From the Table1, the two are with the volume of 150,000 m3and 500 m3, respectively. From the event with 500 m3, there are no stations which record the signal. I suggest the authors can remove this case. From the other case, it's a little strange that the case is with the high volume, but the local magnitude is zero. I think the authors should check the data again or describe the mechanism detail. Final, I also suggest the authors can use different parameters like PGV or envelope area to address this issue.

---

## Referee Comment (RC2) · N. Vouillamoz (Referee) · 2 Aug 2018

**Seismic detection of rockslides at regional scale: Examples from the Eastern Alps and feasibility of kurtosis-based event location by Florian Fuchs et al.**

**Comments referee #2, Naomi Vouillamoz, 02.08.2018**

This paper presents a simple approach to automatically detect, discriminate and locate rockslide events using broadband seismic records at regional scale. The feasibility of the approach is evaluated with 21 rockslide events. The automatic detection applies a recursive STA/LTA coincidence trigger where a minimum number of four stations is needed to declare a detection. It was tested on eightminute continuous data segments around the 21 events, and 19 of the 21 rockslide events (two events being too weak) were successfully detected. For event discrimination between rockslides and earthquakes, 32 earthquakes are selected, which cover the same region and span the same order of magnitudes as the 21 rockslide events. A simple decision criterion based on three seismic features of the earthquakes and rockslide signals enables to successfully discriminate all events. Rockslide events location is performed using a kurtosis-based picker, which permits better picking of the emergent signal onsets and hence an improved and more accurate location of the event. Using this approach, 14 events could be located with deviation ranging from 0.7 to 26 km, 11 events being located with less than 10 km deviation from the true location. A volume-magnitude relation is derived and used to propose a power-law between the seismic amplitude of the signals and the volume of the rockslide event. The authors provide then a detailed discussion about their approach and its potentials for implementation on large continuous seismic datasets. The paper is well written, and well-structured. The approach seems promising. This study should be published with some revisions in the special issue of Earth Surface Dynamics: From process to signal – advancing environmental seismology.

**General comments:**

Should the title of the paper be refined? You propose a simple (and elegant) automatic detection, discrimination and location scheme for rockslide events at regional scale, using low sample-rate broadband data of national seismic networks. It seems the methods could be easily implemented for real-time applications. I have the feeling there might maybe be something even more appealing than the current title, but it's just a feeling (3)...

I don't see the point including the AlpArray Working Group in the author list! Or is this mandatory because you use AlpArray data? This is a small research paper. An acknowledgment and reference as it is already done at the end of the paper is in my opinion enough in that context...

How was the rockslide dataset established? The authors mention in the last paragraph of the discussion section (P12 L9-12) that much knowledge could be gained by merging or cross-checking national event databases over the borders. However, events published for instance by Dammeier et al. (2016) (see Figure 4 of that paper) or the 'famous' August 2017 event of Piz Cengalo (Bondo) which are located in the study area do not figure in the studied dataset. Why?

The discussion part needs to be reworked. The Event Detection section should be discussed in more details with more specific examples on more sensitive algorithms and how one could optimize computational requirements with false alarm rates. The section Kurtosis picker performance and location accuracy could be better structured. I provide more specific comments about that section below. Event discrimination and volume estimation should be split in two individual sections.

**Specific comments:**

P3 L1-2. Please specify better how the dataset was established and the events selected, since between 2007-2017 other events are known (see the above general comment).

P3 L2-3. Out of these 21 events, 17 rockslides have been independently ...; I see 18 events in the Table (only 3 [b]).

P3 L6. "carried out at the Austrian Central Institute..."

P3 L9-10: Please provide a reference for the distance attenuation function used at ZAMG. Specify in the text that  $M_L$  was calculated by ZAMG (instead of only in the Table 1 caption).

P3 Section 2, Dataset: please describe here the 32-earthquake dataset used for event discrimination including a list (and a Table), preferentially providing the same information as for the rockslides.

P4 Table 1. Please provide an event ID reusable in Table 2. Please add a field with the minimal and maximal epicentral distance. Since you use  $M_L$  to derive a Volume-Amplitude scaling relation, an information about the number of amplitude reading used in  $M_L$  estimation would be interesting, especially if different from the number of stations with positive STA/LTA.

P6 L11. For some events a distinct second arrival is visible: How many events exactly? List the events based on event IDs so the reader can go and have a look on the waveform if interested.

P6 L 16. Time is scanned in steps of 2 s (space is missing).

P7 L6-7. For clarity purposes: (1) the Kurtosis...; (2) the ratio between maximum amplitude...; (3) the ratio of the duration...

P8 L1: For clarity purposes: We extract the same three parameters for the earthquake records in order to ...

P8 L22. A local magnitude defined by 4 stations' amplitude reading as it is (?) the case for a couple of rockslide events is not exactly well defined. Moreover,  $M_L$  below 2 is always a bit tricky... What makes you think you are less loosely constrained than other references (3) Please rephrase accordingly.

P9 L22. ... we did not check how many false alarms would be introduced. What a pity! A few tests would have provided very interesting information/benchmark in terms of false alarm rates/data process speed, which is key for real-time implementation.

P10 L2. ... by gravitational mass movements at regional distances.

P10 L2-3. I would rephrase. Eleven of the 14 locatable events in this study could be located within less than 10 km deviation from the true deviation (see Table 2).

P10 L12-14. These two sentences are not very clear. Do you mention the sampling rate and record bandpass as a potential reason for 'bad' locations? You expect better picking accuracy with higher sampling rate records? Please rephrase.

More generally, I would better structure the first paragraph of that section. Provide the reader with ratings. Which parameters are the most influencing?) How much variation did you observed when playing with the kurtosis-based picker? I expect the outliers to have way much influence on a bad location than the optimization of the kurtosis-based piker (see for example Joswig (2008), p 121, box *"Jackknifing explained"* or Vouillamoz et al. (2016), Figure 6).

P10 L24. For those events presenting a very distinct second arrival...

P10 L27. Most of (?) the other events show no clear second onset...

P11 L4. I find 'we demonstrate' a bit ambitious regarding the low statistical significance of the used dataset. We show that rockslides and earthquakes...

P11 L11-14. To my knowledge, machine learning is usually trained on lots of known events, not a few selected known events. Hammer et al. (2013) developed a classifier based on 1 single known events using Hidden Markov Models, however the random forest algorithm of Provost et al. (2016) is trained on hundreds of events. Please rephrase for clarity.

P12 L6. A general drawback of many studies.... this includes also your study. Even if you present more events than other studies, 21 events is still a limited number of events... Please rephrase...

P12 L 18. Again, I find demonstrate a bit too high... We propose a simple approach to search for seismic signatures of rockslides ...

P12 L 20. ... can potentially be reduced. I think greatly is too optimistic and actually, 10 km is not bad at all, given the quality of the onsets and the frequently high gap...

P12 L31. you forgot the final point... 🕹

P13 L29. Team list. Again, I think referring to the AlpArray work group in acknowledgement and in the reference is enough.

Figures:

Figure 1. Please enhance the contrast between the colors of the permanent and the AlpArray stations. Use  $M_{L}$  scaling in the symbology (0-1, 1-2, >2) so the reader can easily recognize the bigger events. Provide lat-lon information or if you don't want to work in a GIS, maybe you could add IDs as label. Please add a Figure 1b, same area and scale, but displaying the earthquakes (also with  $M_{L}$  scaling) so the reader can visually compare the two datasets.

Figure 4. Caption: Use same date format as in the other figures and tables (YYYY-MM-DD).

Figure 5. Caption: Distribution of the three discrimination parameters...

Figure 6. It would be nice to have a word about the outlier at 10^5  $m^3$  and  $M_L\,0.$

**References**

Dammeier, Franziska; Moore, Jeffrey R.; Hammer, Conny; Haslinger, Florian; Loew, Simon (2016): Automatic detection of alpine rockslides in continuous seismic data using hidden Markov models. In *J. Geophys. Res. Earth Surf.* 121 (2), pp. 351–371. DOI: 10.1002/2015JF003647.

Hammer, C.; Ohrnberger, M.; Fah, D. (2013): Classifying seismic waveforms from scratch: a case study in the alpine environment. In *Geophysical Journal International* 192 (1), pp. 425–439. DOI: 10.1093/gji/ggs036.

Joswig, Manfred (2008): Nanoseismic monitoring fills the gap between microseismic networks and passive seismic. special topic, Leveraging Technology. In *first break* 26, pp. 117–124.

Provost, Floriane; Hibert, Clément; Malet, Jean-Philippe (2016): Automatic classification of endogenous landslide seismicity using Random Forest (submitted). In *Geophys. Res. Lett.*

Vouillamoz, Naomi; Wust-Bloch, Gilles Hillel; Abednego, Martinus; Mosar, J. (2016): Optimizing Event Detection and Location in Low-Seismicity Zones: Case Study from Western Switzerland. In *Bull. Seism. Soc. Am. (Bulletin of the Seismological Society of America)*.

---

## Author Comment (AC1) · 19 Sep 2018

**Final Response to Reviews**
*Seismic detection of rockslides at regional scale: Examples from the Eastern Alps and feasibility of kurtosis-based event location*
by F. Fuchs et al.

We thank Naomi Vouillamoz and the anonymous reviewer for their constructive comments. In the following we respond individually to all points raised by both reviewers. A revised version of the manuscript will be uploaded.

We carefully revised the manuscript and hope that it is now in good shape to be accepted for publication in Earth Surface Dynamics.

With kind regards,
Florian Fuchs and co authors

Our replies are structured and color-coded as follows:

**Comments from Referees**
- Authors response
➔ *Modified passages in the manuscript*
* * *
Reviewer 1, Anonymous

**1. The authors mentioned that the purpose of research focuses on automatic location and automatically distinguish earthquake events. But the research uses the known database and construct the algorithms. I would suggest the authors revise the purpose of this research.**

- We use known events to tune and test our algorithms, which then can potentially be applied to find unknown events both in the past and the future. This is future work, however. Here we only suggest several algorithms that could be used. We added this sentence to the introduction to point this out in the very beginning:

➔ *"Here we present a study of 21 rockfalls and rockslides that occurred in or near Austria in the years 2007 to 2017 and were well-recorded by permanent national seismic networks in the Alps during routine earthquake monitoring. We use this dataset of confirmed events to develop and test automatic detection and locating algorithms that could be used to systematically search for additional events in existing and future data. Exploring the feasibility of a country-wide real-time detection scheme for rockfalls, we focus on developing simple automatic location routines and to automatically distinguish such events from regional earthquakes."*

**2. From the reference photo, some rock disaster seems like a free fall event. It has better to define rockslide in the introduction part.**

- We repeatedly state that our events are either rockslides or rockfalls, with the majority being rockslides. The entire documentation about our events is in German language, which doesn't clearly distinguish between rockfalls and rockslides. From just the pictures it is hard to distinguish for some cases. We agree, that e.g. events [2] and [10] look more like rockfalls. However, we are lacking solid proof. Thus we would like to keep our formulations as they are. Anyways, we added a sentence in the discussion section that addresses this issue:

➔ *"For the 21 events in this study we can only estimate the drop mechanism from photographs, which is not always conclusive. While the majority of events would classify as rockslides, some may include a free-fall phase and could rather be regarded as rockfalls (see Table 1)."*

➔ We also updated Table 1, indicating which events might classify as rockfalls rather than rockslides.

**3. In page 5, The STA/LTA method common uses in signal analysis. I think the authors should add some references in this part. Further, why the authors set the different thresholds of trigger-on and trigger-off ratio?**

➔ The trigger-off ratio needs to be lower than the trigger-on ratio in order to have a well-defined end of the event. Otherwise event start time (trigger-on) and event end time (trigger-off) would be identical, or the trigger-off threshold would never be reached after the trigger-on activated. The trigger-off ratio should mark the point were signal amplitudes roughly decayed back to pre-event levels. We believe most readers from seismology will be familiar with this concept and do not further explain it in the manuscript. Anyways, we added a reference which very well describes the principles of an STA/LTA trigger.

➔ "We added this reference: *Trnkoczy, A. Bormann, P. (Ed.) Understanding and parameter setting of STA/LTA trigger algorithm, New Manual of Seismological Observatory Practice 2 (NMSOP2), Deutsches GeoForschungsZentrum GFZ, Potsdam, 2012, 1-20."*

**4. In page 6, Many research support that the P and S wave can't be classified in rockslide/landslide events. From the right part of the Fig.3, the event also looks like containing two parts. The minor event happened first and a major event occurred following. It's a common situation in rockslide event. I suggest the authors carefully check the data again. If this is P and S wave, I think the authors should describe it in detail.**

- We carefully checked those events. Usually, on the closest station, no separate phases or stages can be identified. If the two arrivals were due to two events, we would expect to see them as well at the closest station. More importantly, the two arrivals clearly separate with distance and travel at different velocities. Thus, we are quite certain that this is due to P and S wave propagation, not two separate events. We added a sentence to point this out:

➔ *"We exclude that the two distinct arrivals reflect two separate events, since with increasing distance we observe increasing separation time. In addition, no such separation is visible on the records of the stations closest to the rockslide."*

**5. From the automatic detection, I think it may detect some unknown rockslides, but from authors' data, all rockslides are known events. In advance, I would suggest the authors use different values (like different frequency range) to run the automatic detection. And some deviations of location are quite large. It's a little bit impractical for further application.**

- We believe the manuscript is quite clear about the fact that we try to develop methods based on known events. Here we report on the performance of some simple methods, that might eventually be used for automatic detection in the future. We are not there, yet. In the discussion section we address certain improvements to our methods, including different frequency ranges (and we explain why we couldn't for this study). Concerning the locations: Some are not satisfactory and need further attention. Other, however, are already quite precise and good enough for applications. We do not claim that everything is working optimally, yet, but we want to show up some methods which could be systematically and automatically applied in the future. We think the manuscript is clear about this.

**6. The authors construct eq.(4) to distinguish the earthquakes and rockslides. I would suggest the authors validate the equation with rockslides after February 2017.**

- This is certainly a good suggestions. However, after February 2017 new events have been sparse and we are still collecting events. This is a task for future work.

**7. From the fig 6, two events' local magnitude is zero. If the authors remove these two points, the R2 should be higher. From the Table1, the two are with the volume of 150,000 m3and 500 m3, respectively. From the event with 500 m3, there are no stations which record the signal. I suggest the authors can remove this case. From the other case, it's a little strange that the case is with the high volume, but the local magnitude is zero. I think the authors should check the data again or describe the mechanism detail. Final, I also suggest the authors can use different parameters like PGV or envelope area to address this issue.**

- Following the suggestions from Reviewer 2 we added explanations how the magnitudes were obtained at several places in the manuscript. The Magnitudes are listed like this in the database at ZAMG (Austrian earthquake service). Note, that for small magnitudes such as 0.0 probably only few stations were read for amplitudes (this is also mentioned in the text now). Both 0.0 events do indeed show very low amplitudes, so probably the Magnitudes are correct. For the event in Schwaz, no station activated the STA/LTA detector (thus no entry in Table 1), but the event is visible in the seismic data (very weakly). However, the volume estimate based on reference [9] is probably wrong. Thus, we adjusted Figure 6 and equation 5+6 to show how the result would look like without the 0.0 / 150.000m³ pair.

➔ *Updated Figure 6 + Caption*
➔ *Updated Equations (5) and (6)*
➔ *"Note, that we exclude the data pair ($M_l$=0.0, V=150.000; Schwaz event) since the volume estimate is likely wrong."*

We discuss in section 4 that other authors performed similar studies based on PGV (e.g. Dammeier et al. 2011). Envelope area is an interesting concept (since it includes information about the event duration) and we thank you for pointing this out. We will test this in the future.

Reviewer 2, Naomi Vouillamoz

**Should the title of the paper be refined? You propose a simple (and elegant) automatic detection, discrimination and location scheme for rockslide events at regional scale, using low sample-rate broadband data of national seismic networks. It seems the methods could be easily implemented for real-time applications. I have the feeling there might maybe be something even more appealing than the current title, but it's just a feeling.**

- Fair point, but we didn't really come up with a better title ...

**I don't see the point including the AlpArray Working Group in the author list! Or is this mandatory because you use AlpArray data? This is a small research paper. An acknowledgment and reference as it is already done at the end of the paper is in my opinion enough in that context...**

- Including the AlpArray Working Group in the author list as well as listing all names in the acknowledgment section is a requirement for all AlpArray-based publications that the AlpArray group agreed on. Thus, even though we use little data from the AlpArray network, we have to leave it as it is.

**How was the rockslide dataset established? The authors mention in the last paragraph of the discussion section (P12 L9-12) that much knowledge could be gained by merging or cross-checking national event databases over the borders. However, events published for instance by Dammeier et al. (2016) (see Figure 4 of that paper) or the 'famous' August 2017 event of Piz Cengalo (Bondo) which are located in the study area do not figure in the studied dataset. Why?**

- The dataset was compiled over the years at ZAMG, Austria, and comprises strong events that were recognized during routine earthquake monitoring, or were found after ZAMG was alerted of a strong event by the geological Service of Austria (GBA). Thus the dataset naturally contains events only from Austria and South-Tyrol, with two exceptions from Switzerland (these are also referenced in Dammeier et al. 2016). Very recent events from 2017 and 2018 are not included as we stopped the work at this point and started writing it up. We currently work on the August 2017 Piz Cengalo slide which will be subject of another publication. We added the following sentences to Section 2, to clarify how the dataset was compiled:

➔ *"The event database was compiled by the Austrian earthquake service and focuses on rockslides and rockfalls from Austria and South-Tyrol (Italy). These events were manually detected and classified during routine earthquake monitoring by the Austrian earthquake service (Central Institute for Meteorology and Geodynamics, ZAMG), and verified in cooperation with the Austrian Geological Service (GBA). We additionally include two large-scale rockslides that occurred in Switzerland, but were also detected by the Austrian colleagues and assigned a magnitude."*

**The discussion part needs to be reworked. The Event Detection section should be discussed in more details with more specific examples on more sensitive algorithms and how one could optimize computational requirements with false alarm rates. The section Kurtosis picker performance and location accuracy could be better structured. I provide more specific comments about that section below. Event discrimination and volume estimation should be split in two individual sections.**

- We reworked parts of the discussion section following this suggestion. Even detection now includes more explanation on STA/LTA false alarms and other more sensitive algorithms. We added several more examples on alternative detectors and included references. We reworked the Kurtosis picker section according to the comments below (see below). We split event discrimination and volume estimation in two separate sections.

➔ *"However, the STA/LTA triggering threshold level of 4.0 used in this study is commonly used for averagely quiet sites (Trnkoczy 2012). Increasing the number of stations needed for a positive result can in case be used to lower the false alarm rate."*

➔ *"Manconi et al. 2016 report that the predictive multi-band detector FilterPicker (Lomax et al 2012) is suitable to detect and phase-pick emergent seismic signals of rockslides. Lassie is a stack-and-delay based coherence detector to find and locate events at the same time (Lopez et al 2017, Heimann et al. 2018) and may also be applicable to rockslide signals. Soubestre et al. 2018 demonstrate how coherent volcanic tremor signals can be detected and classified on a regional seismic network based on network covariance matrices. Since rockslide signals in several aspects resemble tremor signals (emergent onset, long duration, frequency content) this concept might as well be applicable to rockslide detection. Template matching and subspace detectors (Maceira et al 2010) are commonly used for earthquake and tremor detection, but we speculate that such methods may not be suitable for rockslide detection, as for every event waveforms are highly individual because of the complexity and variability in source mechanisms."*

➔ *We split Section "Event discrimination & volume estimation" into two separate ones.*

**P3 L1-2. Please specify better how the dataset was established and the events selected, since between 2007-2017 other events are known (see the above general comment)**

- Done, please see comment above

**P3 L2-3. Out of these 21 events, 17 rockslides have been independently …; I see 18 events in the Table (only 3 [b]).**

- We agree that listing Mellental 3x was confusing. We now summarize all three Mellental events into a single one and adjusted the caption accordingly. Still, we are left with 19 events.

➔ *Table 1 Caption addition: "The Mellental event occurred in three stages. The magnitude refers to the first event in the sequence. The volume estimates the total mass loss over all stages."*

*Accordingly, we corrected the number of events from 21 to 19 throughout the manuscript.*

**P3 L6. "carried out at the Austrian Central Institute…"**

- This paragraph changed, the sentence no longer exists. See comments above.

**P3 L9-10: Please provide a reference for the distance attenuation function used at ZAMG. Specify in the text that ML was calculated by ZAMG (instead of only in the Table 1 caption)**

- We added passages pointing out that Ml was calculated by ZAMG. Unfortunately no reference exists that explains the distance attenuation used at ZAMG.

**P3 Section 2, Dataset: please describe here the 32-earthquake dataset used for event discrimination including a list (and a Table), preferentially providing the same information as for the rockslides.**

- Done. However, instead of expanding Section 2 we added Supplemental Online Material to the manuscript that describes how the earthquake dataset was obtained. There we also show a map of the earthquake dataset (Fig. S1, similar for Figure 1) and a table listing the same information as for the rockslides (Table S1, similar to Table 1). We also added additional text to 3 to refer to the new Supplemental Material.

  ➔ *New supplemental online material that describes how the earthquake dataset was obtained.*
  ➔ *Added in Section 3 (Discrimintation): "(see Fig. S1 and Table S1 and the Supplemental Online Material for details))"*

**P4 Table 1. Please provide an event ID reusable in Table 2. Please add a field with the minimal and maximal epicentral distance. Since you use ML to derive a Volume-Amplitude scaling relation, an information about the number of amplitude reading used in ML estimation would be interesting, especially if different from the number of stations with positive STA/LTA.**

- We believe that the event name serves well to find any event in Table 2. Since Table 2 is sorted by location deviation, any numbering scheme would be anyways mixed between Table 1 and Table 2. Unfortunately the number of amplitude readings for the determination of Ml is not available to us. We added the min and max epicentral distance to the table.

  ➔ *Added new column (Distance / km) to Table 1. Added to caption: "The distance column indicates the minimum and maximum distance from the events for these stations."*

**P6 L11. For some events a distinct second arrival is visible: How many events exactly? List the events based on event IDs so the reader can go and have a look on the waveform if interested.**

- Done, we added the number and listed the events.

  ➔ *"For eight events (Einserkofel, Hochwand, Gamsgrube, Trins, Stubaital, Dobratsch, Mellental, Zwölferkofel) a distinct second ..."*

**P6 L 16. Time is scanned in steps of 2 s (space is missing).**

- Fixed

**P7 L6-7. For clarity purposes: (1) the Kurtosis…; (2) the ratio between maximum amplitude…; (3) the ratio of the duration…**

- Done

**P8 L1: For clarity purposes: We extract the same three parameters for the earthquake records in order to …**

- Done

**P8 L22. A local magnitude defined by 4 stations' amplitude reading as it is (?) the case for a couple of rockslide events is not exactly well defined. Moreover, ML below 2 is always a bit tricky… What makes you think you are less loosely constrained than other references ? Please rephrase accordingly.**

- In fact, we don't know quality of the Magnitude estimates by ZAMG. We removed "well defined" and instead point out, that Magnitudes were assigned by ZAMG during monitoring. To make readers aware of potentially uncertain Magnitudes we added the following sentence to the discussion section. However, we cannot give any bounds on this uncertainty.

➔ *"Note, however, that apart from the volume estimate also the local Magnitude may not be very well-defined, especially for low-magnitude ($M_l < 2$) events with only few amplitude readings available."*

**P9 L22. … we did not check how many false alarms would be introduced. What a pity! A few tests would have provided very interesting information/benchmark in terms of false alarm rates/data process speed, which is key for real-time implementation.**

- Yes, we agree that this would be crucial for real-time applications. However, this was out of the scope of the current study. We suspect that because of the relatively high detector activation threshold and the requirement of minimum four activated stations, the false alarm rate would be acceptable. We refrain from speculating in the manuscript though and would leave the statement as it is. We will address this issue in a planed publication that is based on the same procedure.

**P10 L2. … by gravitational mass movements at regional distances.**

- Done

**P10 L2-3. I would rephrase. Eleven of the 14 locatable events in this study could be located within less than 10 km deviation from the true deviation (see Table 2).**

- We would prefer to keep it as it is.

**P10 L12-14. These two sentences are not very clear. Do you mention the sampling rate and record bandpass as a potential reason for 'bad' locations? You expect better picking accuracy with higher sampling rate records? Please rephrase.**

- We reformulated these sentences and hope they are more clear now.

➔ *"Future work should include all three components of the seismic record and use different narrow frequency bands for comparison, as suggested by (...). We expect that evaluating the kurtosis pick among different frequency bands would suppress outliers (due to noise) and thus make the onset determination more robust and precise. Yet, in this study - due to low sampling rate for older records - we could not extend the processing to higher frequencies. Lower frequencies are very weak in amplitude or absent for almost all rockslides in this study. This is in line with observations from several} other studies that report the 1--5 Hz frequency range as the dominant one for regional seismic records of rockslides (...)."*

**More generally, I would better structure the first paragraph of that section. Provide the reader with ratings. Which parameters are the most influencing?) How much variation did you observed when playing with the kurtosis-based picker? I expect the outliers to have way much influence on a bad location than the optimization of the kurtosis-based piker (see for example Joswig (2008), p 121, box "Jackknifing explained" or Vouillamoz et al. (2016), Figure 6).**

- We do mention the possible variation of location results ("few kilometers") and reasons ("moving window length") in the text. We added the corner frequencies of the bandpass filter as one of the influencing parameters. We agree that outliers have a huge impact for some events and added sentences pointing this out. We do acknowledge that a systematic search for variability is a convenient way to explore uncertainties and parameter sensitivity. However, we feel that our limited dataset and the currently simple processing are not worth the effort. We agree that this should be done ones the processing scheme will be applied on larger scale to find new events.

➔ *"We do in fact observe that the location results currently lack robustness and may change by few kilometers when certain parameters of the kurtosis picker (e.g. the length of the moving window; bandpass filter corner frequencies) are adjusted."*

➔ *"Additionally, we did not implement automatic outlier handling at this stage. Several of the bad locations listed in Table 2 can be explained by strong outliers in the kurtosis picks due to noise. We expect that picking accuracy can be greatly improved if measures are taken to make the kurtosis picker more robust and to exclude outliers."*

**P10 L24. For those events presenting a very distinct second arrival…**

- We kept the sentence as it is, but listed again the events in question.

**P10 L27. Most of (?) the other events show no clear second onset…**

- Changed to:

➔ *"The majority of events …*

**P11 L4. I find 'we demonstrate' a bit ambitious regarding the low statistical significance of the used dataset. We show that rockslides and earthquakes…**

- Agreed and changed

**P11 L11-14. To my knowledge, machine learning is usually trained on lots of known events, not a few selected known events. Hammer et al. (2013) developed a classifier based on 1 single known events using Hidden Markov Models, however the random forest algorithm of Provost et al. (2016) is trained on hundreds of events. Please rephrase for clarity.**

- We expanded this paragraph as suggested and give more information about the number of training events needed. It now reads:

➔ *"Dammeier et al. 2016 demonstrate how a single training event can be used to scan continuous data for rockslides based on Hiden Markov Models"*

➔ *"Random forest classifiers work more reliable the more training events are available. Recent studies demonstrate that sensitivities higher than 85% can be achieved if just 10% of the events inside a dataset are used to train the algorithm (Provost et al. 2017, Hibert et al. 2018). In the work of Provost et al. 2017 this corresponds to 20-40 training events per event category, which is in the same order of magnitude as the number of events in this study, suggesting that these could be sufficient to screen larger datasets."*

**P12 L6. A general drawback of many studies…. this includes also your study. Even if you present more events than other studies, 21 events is still a limited number of events… Please rephrase…**

- We now mention that this explicitly includes our own study.

➔ *"A general drawback of many studies \textcolor{red}{(including this one)} that aim to ..."*

**P12 L 18. Again, I find demonstrate a bit too high… We propose a simple approach to search for seismic signatures of rockslides …**

- We changed this, it now reads:

➔ *"We have outlined simple methods how to search for ..."*

**P12 L 20. … can potentially be reduced. I think greatly is too optimistic and actually, 10 km is not bad at all, given the quality of the onsets and the frequently high gap...**

- We would like to stay a bit optimistic here. Some examples from our study show that if there are no outliers the location accuracy can be brought down to few (1-3) km which is of similar quality as for earthquakes. Anyways, we changed "*greatly*" to "*further*"

**P12 L31. you forgot the final point…**

- Well spotted, shame on us! :) Fixed of course …

**P13 L29. Team list. Again, I think referring to the AlpArray work group in acknowledgement and in the reference is enough.**

- See first comment. AlpArray rules require us to do this.

**Figure 1. Please enhance the contrast between the colors of the permanent and the AlpArray stations. Use ML scaling in the symbology (0-1, 1-2, >2) so the reader can easily recognize the bigger events. Provide lat-lon information or if you don't want to work in a GIS, maybe you could add IDs as label. Please add a Figure 1b, same area and scale, but displaying the earthquakes (also with ML scaling) so the reader can visually compare the two datasets**

- Temporary stations are now filled markers and colored black. This should do the job. We added Lat/Lon markers. We tested maps with marker sizes scaler with magnitude, but it did not increase the readability. Thus, we would like to keep it as it is. We added a map for the earthquakes as Supplemental Material.

➔ *Figure 1 updated*

**Figure 4. Caption: Use same date format as in the other figures and tables (YYYY-MM-DD).**

- Done

**Figure 5. Caption: Distribution of the three discrimination parameters…**

- Done

**Figure 6. It would be nice to have a word about the outlier at 10^5 m3 and ML 0.**

- See response to reviewer #1. We updated Figure 6 and added passages to the text.

**Additional changes:**

- *Corrected number of events (landslides 21→ 19; earthquakes 32 → 31) throughout the manuscript.*
- *Updated event reference [15]: Link was deprecated. Was uploaded the original report to our server.*
- *Updated event reference [17]: Link was outdated and no longer working. We added a new one*
- *Added references:*

*Trnkoczy, A. Bormann, P. (Ed.) Understanding and parameter setting of STA/LTA trigger algorithm New Manual of Seismological Observatory Practice 2 (NMSOP2), Deutsches GeoForschungsZentrum GFZ, Potsdam, 2012, 1-20*

*Fuchs, F.; Kolínský, P.; Gröschl, G.; Apoloner, M.-T.; Qorbani, E.; Schneider, F. & Bokelmann,  G. Site selection for a countrywide temporary network in Austria: noise analysis and preliminary performance Advances In Geosciences, 2015, 41, 25-33*

*Heimann, S., Matos, C., Cesca, S., Rio, I., and Custodia, S.: Lassie: A versatile tool to detect and locate seismic activity, in preparation; Note: interested users to preview Lassie can write to: sebastian.heimann@gfz-potsdam.de, 2018*

*Lomax, A., Satriano, C., and Vassallo, M.: Automatic Picker Developments and Optimization: FilterPicker - a Robust, Broadband Picker for Real-Time Seismic Monitoring and Earthquake Early Warning, Seismological Research Letters, 83, 531–540*

Lopez Comino, J. A., Heimann, S., Cesca, S., Milkereit, C., Dahm, T., and Zang, A.: Automated Full Waveform Detectionand Location Algorithm of Acoustic Emissions from Hydraulic Fracturing Experiment, Procedia Engineering, 191, 697–702

Maceira, M., Rowe, C. A., Beroza, G., and Anderson, D.: Identification of low-frequency earthquakes in non-volcanic tremor using the subspace detector method, Geophysical Research Letters, 37, L06 303

Soubestre, J., Shapiro, N. M., Seydoux, L., de Rosny, J., Droznin, D. V., Droznina, S. Y., Senyukov, S. L., and Gordeev, E. I.: Network-Based Detection and Classification of Seismovolcanic Tremors: Example From the Klyuchevskoy Volcanic Group in Kamchatka, Journal of Geophysical Research: Solid Earth, 123, 564–582

---

## Author Response (AR2)

Dear Editors,

we thank you for your positive feedback and for accepting our work for publication in the ESurf Special Issue on Environmental Seismology.

For this final version we adopted the volume column in Table 1, according to your suggestions. All values are now given in thousands of cubic meters. Unfortunately we do not have proper uncertainty estimates for any of the events, but whenever possible we state the volume range as taken from the respective references.

Note, that we also updated event reference [14] (Mellental). The report is now available for download and we updated the corresponding link.

Thank you very much and with kind regards,

Florian Fuchs and co-authors